# Combining CD3/GD2 bispecific T cell engager with human Vγ9Vδ2 T cells facilitates neuroblastoma cell targeting and killing *in vitro*

Kuntida Kitidee[1], Sumet Amonyingcharoen[2], Sarinthip Preedagasamzin[2], Korakot Atjanasuppat[2], Piamsiri Sawaisorn[3], Pornprapa Srimorkun[4], Sawang Petvises[4], Wanpen Chaicumpa[5], Suparerk Borwornpinyo[6,7], Usanarat Anurathapan[2], Suradej Hongeng[2]*

**1** Center for Research Innovation and Biomedical Informatics, Faculty of Medical Technology, Mahidol University, Salaya, Nakhon Pathom, Thailand, **2** Department of Pediatrics, Faculty of Medicine Ramathibodi Hospital, Mahidol University, Bangkok, Thailand, **3** Department of Clinical Microscopy, Faculty of Medical Technology, Mahidol University, Nakhon Pathom, Thailand, **4** Department of Medical Technology, Faculty of Allied Health Sciences, Thammasat University, Pathum Thani, Thailand, **5** Center of Research Excellence in Therapeutic Proteins and Antibody Engineering, Department of Parasitology, Faculty of Medicine Siriraj Hospital, Mahidol University, Bangkok, Thailand, **6** Department of Biotechnology, Faculty of Science, Mahidol University, Bangkok, Thailand, **7** Excellent Center for Drug Discovery, Faculty of Science, Mahidol University, Bangkok, Thailand

* suradej.hon@mahidol.ac.th

## Abstract

Cancer immunotherapy, particularly T cell–based therapies, is considered to have strong potential for treating various types of cancer. A promising approach that has emerged is the use of γδ T cell-based strategies for cancer treatment. Neuroblastoma (NB), a solid tumor frequently found in childhood, is one of the more intriguing targets for immunotherapy. In this study, we report an alternative immunotherapy method for treating neuroblastoma by combining bispecific antibody with human Vγ9Vδ2 T cells. Initially, we screened for human scFv against CD3 epsilon using phage panning technology. Human scFv CD3 clone 18 demonstrated the highest ability to bind CD3 epsilon in an indirect ELISA assay. Consequently, we selected human scFv CD3 clone 18 to create a bispecific T cell engager antibody targeting both CD3 and disialoganglioside (GD2), called CD3/GD2 BiTE. This bispecific antibody was composed of human scFv CD3 clone 18 (VH-VL) and mouse scFv GD2 (VL-VH), linked by a flexible peptide linker. The interleukin-2 signal sequence and polyhistidine tag were added at the N- and C-termini for protein secretion and purification, respectively. CD3/GD2 BiTE was transiently produced in a mammalian cell expression system, which provided both high yield and quality. The CD3/GD2 BiTE folded naturally into a compact monomeric structure. Cell-based binding activity assays demonstrated that CD3/GD2 BiTE specifically binds to its target antigens on CD3-positive Jurkat cells and GD2-positive SH-SY5Y cells, but did not react with CD3-negative Raji cells and GD2-negative SK-N-SH cells. In subsequent *in vitro* experiments, the cytotoxicity of

**Data availability statement:** All relevant data are within the paper and its Supporting Information files.

**Funding:** This work was supported by the Research Chair Grant from the National Science and Technology Development Agency, Thailand (FDA-CO-2559-3325-TH) to Suradej Hongeng and the Fundamental Fund from Thailand Science Research and Innovation (FRB650007/0185) to Usanarat Anurathapan. In this study, the funders had no role in study design, data collection and analysis, decision to publish, or preparation of the manuscript.

**Competing interests:** The authors have declared that no competing interests exist.

CD3/GD2 BiTE combined with human Vγ9Vδ2 T cells against neuroblastoma cells was evaluated. Human Vγ9Vδ2 T cells were primed with CD3/GD2 BiTE to improve the binding specificity and avidity against neuroblastoma cell lines before adding into SH-SY5Y cells. At concentrations of 180 and 360 nM, the CD3/GD2 BiTE significantly enhanced the killing ability of human Vγ9Vδ2 T cells against SH-SY5Y cells at an E:T ratio of 1:1. Moreover, CD3/GD BiTE armed with human Vγ9Vδ2 T cells enabled the killing of neuroblastoma cells using five- to ten-times fewer effector cells. The combination of CD3/GD2 BiTE and human Vγ9Vδ2 T cells also exhibited cytotoxic activity against a three-dimensional tumor spheroid model of SH-SY5Y GFP at an E:T ratio of 1:1. Consequently, CD3/GD2 BiTE enhances tumor-targeting and cytotoxic abilities of human Vγ9Vδ2 T cells against neuroblastoma cells in both two-dimensional and three-dimensional cell cultures. These results suggest that the combination of CD3/GD2 BiTE and human Vγ9Vδ2 T cells could represent an alternative immunotherapy strategy for treating neuroblastoma patients in the future.

## Introduction

Neuroblastoma (NB) is the most common extracranial solid tumor in children. This pediatric tumor originates from the developing peripheral sympathetic nervous system, particularly from neural crest cells, and contributes to childhood cancer mortality. NB arises in many areas including the abdomen, chest, neck, and near the spine, with the adrenal glands being the most common site. NB exhibits remarkable genetic heterogeneity, involving several chromosomal or gene alterations as well as the uncontrolled expression of oncogenes and tumor suppressor genes, which drive tumor initiation and promote disease progression [1]. It is the most common malignancy in infants, accounting for 6% of all childhood cancers and 15% of childhood cancer deaths [2–4]. The 5-year relative survival rate for children with low-risk NB exceeds 95%, while those with intermediate-risk NB demonstrate survival rates between 90-95%. However, the 5-year relative survival rate of children with high-risk NB is less than 50% [5,6]. A multidisciplinary treatment approach, chemotherapy, surgery, radiation therapy, autologous stem cell transplantation, and immunotherapy, has improved overall survival rates, with the recent 5-year survival rate reaching 79%. Despite these advancements, high-risk neuroblastoma patients, who often show low initial response and have a poor prognosis, still face a survival rate below 50%, presenting a significant challenge [7,8]. Therefore, NB remains a critical health issue in pediatric oncology, and alternative therapeutic strategies are urgently needed to improve treatment outcomes for high-risk NB patients.

Disialoganglioside 2 (GD2) is a cell surface glycosphingolipid that has limited expression in normal tissues but is overexpressed in a wide range of tumors originating from neuroectoderm-derived cells, such as neuroblastoma, and melanoma [9]. High levels of GD2 are correlated with the rapid progression of NB, and lower survival rates are associated with elevated GD2 levels in blood circulation [10]. Therefore, GD2 is considered a tumor-associated antigen and an important therapeutic

target for NB. Currently, a GD2-specific antibody has been approved for clinical use in NB patients. Treatment of high-risk NB with dinutuximab (a monoclonal antibody against GD2), either alone or in combination with cytokines such as granulocyte-macrophage colony-stimulating factor (GM-CSF), interleukin-2 (IL-2), and 13-cis-retinoic acid (RA), has been shown to improve 2-year event-free survival and overall survival compared to standard treatment [11]. Despite the potential suitability of the anti-GD2 monoclonal antibody for NB immunotherapy, at least 30% of NB patients still experience relapse [12]. Enhancing immunotherapeutic strategies by increasing the efficacy and reducing the toxicity of these specific antibodies is crucial for improving outcomes for NB patients.

Immunotherapy has shown remarkable success in treating several types of cancer. Bispecific T-cell engagers (BiTEs) are a promising immunotherapeutic approach designed specifically for cancer treatment. BiTEs are a class of artificial bispecific antibodies that simultaneously engage cytotoxic T-cells and tumor cells through tumor-associated antigens (TAAs), resulting in tumor cell destruction [13]. These engineered heterodimeric antibodies consist of two different antigen-binding sites within a single molecule, one site targets a TAA, while the other binds to the CD3 molecule on the T-cell surface [14]. By retargeting T-cells to attack tumor cells, BiTEs have demonstrated therapeutic efficacy. The most well-known FDA-approved drug in this class is blinatumomab, a single-chain bispecific antibody that binds to CD3 on T-cells and CD19 on B-cell leukemia, thereby recruiting activated T-cells to kill malignant B-cell precursors [15]. The original blinatumomab molecule is currently being remodeled to target other tumor antigens, such as CD33 [16], epidermal growth factor receptor variant III (EGFRvIII) [17], and fibroblast growth factor-inducible 14 (Fn14) [18]. Recently, the safety and efficacy of BiTEs targeting CD3 molecules and TAAs have been explored in more than 100 clinical trials. However, several key challenges remain with BiTEs, including the recruitment of counterproductive CD3+ T-cell subsets, the release of systemic cytokines, the expansion of immune checkpoint molecules, the presence of an immunosuppressive tumor microenvironment, tumor antigen loss or escape, on-target off-tumor toxicity, and suboptimal potency [19].

Immunotherapy with unconventional T-cells, such as γδ T cells, is one of the emerging strategies for cancer treatment. γδ T-cells are a subset of T lymphocytes involved in immune surveillance against infections and tumors. The predominant type of γδ T-cell in human peripheral blood is the Vγ9Vδ2 T-cell, which accounts for approximately 1–5% of all T-cells [20]. Vγ9Vδ2 T-cells can induce apoptosis across a broad spectrum of cancer cells by activating MHC-independent recognition of non-peptide prenyl pyrophosphate antigens, which are upregulated on malignant cells [21]. Consequently, Vγ9Vδ2 T-cells are capable of detecting tumor cells and initiating anticancer responses, including cytokine production and cytotoxicity. However, safety issues have been reported in Vγ9Vδ2 T-cells-based immunotherapy for hematologic malignancies [22,23]. The concept of using γδ T-cells in combination with bispecific antibodies for cancer immunotherapy has been proposed for more than three decades [24]. For example, combining bispecific antibodies (such as anti-HER2 and anti-Vγ9) with the transferred Vγ9Vδ2 T-cells has been shown to trigger γδ T cell-mediated tumor cell killing in pancreatic ductal adenocarcinoma cell lines [25]. Recent studies have demonstrated that the integrated application of bispecific antibodies with γδ T-cells can be effective in treating various cancers, including myeloid leukemia, lung cancer, and breast cancer [26–28]. However, the combination of bispecific antibodies and Vγ9Vδ2 T-cell specifically for targeting neuroblastoma has not yet been investigated in clinical studies or trials.

In this study, we isolated human single-chain variable fragments (scFv) that specifically bind to CD3 epsilon from an M13 phage library displaying naïve human scFv fragments, using phage display biopanning. The human scFv CD3 clone 18, which demonstrated the highest binding activity against CD3 epsilon in an indirect ELISA, was selected for conducting a bispecific T-cell engager. The chimeric bispecific antibody, termed CD3/GD2 BiTE, was created by combining human scFv CD3 clone 18 with mouse scFv GD2. The CD3/GD2 BiTE protein was produced transiently using a mammalian cell expression system. This CD3/GD2 BiTE was able to simultaneously bind to the CD3 receptor on T-cells and the tumor-associated GD2 antigen on neuroblastoma cells. Then, CD3/GD2 BiTE was combined with activated Vγ9Vδ2 T cells to the binding specificity and avidity against neuroblastoma cell lines. *In vitro* cytotoxicity assays of CD3/GD2 BiTE combined with human Vγ9Vδ2 T-cells revealed enhanced tumor targeting and cytotoxic effects on neuroblastoma cell

lines in both two-dimensional monolayer and three-dimensional cultures. These findings suggest an alternative immuno-therapy strategy to improve the treatment of neuroblastoma in the future.

## Materials and methods

### Cell culture

The human neuroblastoma cell lines, including SH-SY5Y and SK-N-SH, were obtained from American Type Culture Collection (ATCC) and cultured in high-glucose DMEM medium (Hyclone, Cytiva, USA). A green fluorescent SH-SY5Y cell line (SH-SY5Y GFP) was established through stable transfection and cultured in high-glucose DMEM medium. Jurkat cells, a T-cell leukemia cell line, and Raji cells, a B-cell lymphoma line, were cultured in RPMI-1640 medium (Hyclone, Cytiva, USA). Human embryonic kidney HEK 293T cells, used for CD3/GD2 BiTE production, were cultured in high-glucose DMEM medium. All media were supplemented with 10% FBS (Gibco, CA, USA), 100 units/ml penicillin, and 100 µg/ml streptomycin (Gibco, Carlsbad, CA). All cell lines were incubated at 37°C in a 5% $CO_2$ and 70-80% humidified atmosphere.

### *In vitro* expansion of human Vγ9Vδ2 T lymphocytes

The study was conducted in accordance with the Declaration of Helsinki. Participants were informed about the research project and provided their consent by signing an informed consent form. Blood specimens of healthy participants were collected at Ramathibodi Hospital, Mahidol University, during October 1, 2021, to October 30, 2022. The collection of healthy blood samples was approved by the Ethics Committee on Research Involving Human Subjects of Ramathibodi Hospital (MURA2020/879). All procedures were performed following relevant guidelines and regulations. Samples were analyzed anonymously to ensure confidentiality and protect participants' privacy rights throughout the study. Human peripheral blood mononuclear cells (PBMCs) were isolated from heparinized blood samples using density gradient centrifugation (Lymphoprep, Oslo, Norway). PBMCs were adjusted to a final concentration at $1 \times 10^6$ cells/ml in complete RPMI 1640 medium, supplemented with 10% FBS, 100 units/ml penicillin, 100 µg/ml streptomycin, 2 mM L-glutamine (Gibco, Carlsbad, CA), 5 µM zoledronic acid (ZOL, ZOLennic, Bangkok, Thailand), and 50 U/ml recombinant human interleukin-2 (PeproTech, Rocky Hill, NJ) in 6-well flat-bottom plates (Costar, Cambridge, USA). Cells were cultured at 37°C in 5% $CO_2$ for 14 days, with the medium supplemented with 50 U/ml of IL-2 being replaced every 3 days.

### Characterization of human Vγ9Vδ2 T lymphocytes by multiparametric immunophenotyping

The expression of cell surface markers on activated Vγ9Vδ2 T lymphocytes was analyzed using flow cytometric antibody staining. Prior to staining, an Fc block solution (BioLegend, USA) was added to the cells to prevent non-specific Fc receptor binding. 7-Aminoactinomycin D (7-AAD) (eBioscience, CA, USA) was used to differentiate live cells from dead cells. Antibodies specific for CD45-Pacific Blue (Clone: 2D1), CD3-APC (clone: OKT3), Vγ9-FITC (Clone: B3), and Vδ2-PE (Clone: B3) were obtained from BioLegend (San Gabriel, CA, USA). Multiparameter flow cytometric analysis was performed using a FACSLyric flow cytometer (BD Biosciences, USA).

### Phage display biopanning of human scFv antibody

An M13 phage library containing a naïve human scFvs gene insert in the phage genome and displaying human scFv fragments on the phage surface was used for biopanning [29]. Phage display biopanning for naïve human scFv against CD3 was performed using recombinant CD3 epsilon (Sigma-Aldrich, USA) as the target protein. Recombinant CD3 epsilon, at a concentration of 0.5 µg, was coated onto 8 ELISA wells (EIA/RIA high protein binding affinity, Costar, USA) and incubated overnight at 4°C. The pre-coated wells were blocked with a blocking buffer (2% BSA in TBS) for 1 h at room temperature, followed by washing with a washing buffer (0.05% Tween 20 in TBS). Subsequently, the phage-displayed naïve

human scFv library was added and incubated for 1 h at room temperature. Unbound phages were removed by washing with 300 µl of washing buffer 10 times. Antigen-bound phages were then eluted with 50 µL of 0.5 M HCl-glycine at pH 2.2 and immediately neutralized with 3 µL of 2 M Tris-base. Afterward, 200 µl of log-phase *E. coli* strain HB2151 was added to the well and incubated for 20 min at 25°C to allow phage transfection into the *E. coli*. The phage-transformed *E. coli* HB2151 were plated on LB agar containing ampicillin (Bio Basic Inc, Canada) and incubated overnight at 37°C. Ninety colonies were picked and screened for the insertion of human scFv genes by colony PCR technique. The recombinant *E. coli* HB2151 clones containing the human scFv gene were cultured and induced for protein expression using IPTG at 30 °C overnight. Total cell lysates from each clone were extracted and tested for the binding activity of soluble human scFv against recombinant CD3 epsilon by indirect ELISA using an HRP-conjugated anti-E tag (GE Healthcare, UK). After ELISA screening, positive clones demonstrating the highest signals were selected and further analyzed for protein expression, binding activity, and nucleotide sequence.

## Subcellular fractionation

The recombinant HB2151 *E. coli* clones containing the human scFv gene were cultured in 10 mL of LB broth containing ampicillin (Bio Basic Inc, Canada) and induced for protein expression using IPTG at 30°C overnight. The recombinant *E. coli* clones were then pelleted by centrifugation at 1,811 × g for 10 minutes and washed once with cold PBS. The washed pellet was resuspended gently in 1 mL of fractionation buffer (30 mM Tris-HCl pH 8.0, 20% Sucrose, 1 mM Na$_2$EDTA) and incubated at room temperature for 10 minutes. The cells were centrifuged at 4,472 × g for 10 minutes, and the supernatant was discarded. The pellet was gently resuspended in 266 µL of 5 mM MgSO$_4$ and incubated on ice for 10 minutes. After centrifugation at 7,558 × g for 10 minutes, the supernatant was collected as the periplasmic fraction. The remaining cell pellet was resuspended with 266 µl of PBS and lysed by sonication. The cell lysate was centrifuged at 7,558 × g for 10 minutes, and the supernatant was collected as the cytoplasmic fraction. Recombinant HB2151 *E. coli* without the scFv gene was used to prepare the total cell lysate, periplasmic fraction, and cytoplasmic fraction as a negative protein control.

## Indirect ELISA

Microtiter plates (Greiner CELLSTAR, Merck, Germany) were coated with 0.5 µg of recombinant CD3 epsilon and incubated overnight at 4 ºC. Bovine serum albumin (BSA) (Calbiochem, Merck, Germany) at 0.5 µg was used as an irrelevant antigen. The coated wells were washed with 200 µl of PBST and blocked with 200 µl of 5% skim milk in PBS for 1 h at 37ºC. After three washes, 500 µg/ml of human scFv CD3 protein from total cell lysate, periplasmic fraction, or cytoplasmic fraction was added to the wells and incubated at 37ºC for 1 h. Unbound proteins were removed by washing 3 times with washing buffer. HRP-conjugated anti-E tag (GE Healthcare, UK) was added to the wells to detect scFvs bound to recombinant CD3 epsilon. The wells were incubated at 37ºC for 1 h and then washed 3 times with washing buffer. Finally, 50 µl of ABTS (2,2'-azino-bis (3-ethylbenzothiazoline-6-sulphonic acid)) substrate (Vector Laboratories, USA) was added. The absorbance at 405 nm was measured using a microplate reader (BioTek Synergy HTX, USA). Recombinant HB2151 *E. coli* without the scFv gene was used to prepare total cell lysate, periplasmic fraction, or cytoplasmic fraction as a negative protein control. The absorbance of soluble human scFv CD3 against CD3 epsilon was analyzed by subtracting the absorbance of the negative protein in CD3 epsilon, followed by the absorbance of soluble human scFv CD3 in BSA-coated well, respectively.

## Generation of bispecific antibodies construct

The sequence of single-chain variable fragment against CD3 (human scFv-CD3 clone 18), derived from phage display selection, and the sequence of mouse single-chain variable fragment against GD2 (mouse scFv-GD2), kindly provided by Professor Malcolm K. Brenner (Baylor College of Medicine, Texas), were obtained [30,31]. The mouse scFv-GD2

originated from the well-known GD2-specific antibodies, 14G2a [32]. The gene encoding the bispecific antibody, consisting of human scFv-CD3 linked with mouse scFv-GD2 (CD3/GD2 BiTE), was designed and optimized for codon usage in the mammalian cell expression system. The mammalian cell expression vector, pcDNA3.1(+), containing CD3/GD2 BiTE (pcDNA3.1-CD3/GD2 BiTE), was constructed by Genscript Biotech Corporation (NJ, USA).

## Expression and purification of bispecific antibodies

The expression vector, pcDNA3.1-CD3/GD2 BiTE, was transfected into the HEK 293T cell line using Lipofectamine 3000 reagent (Invitrogen, Thermo Fisher Scientific, USA). After transfection, the cells were cultured for 72 h, and the culture medium was collected to purify the bispecific antibody protein using Ni-NTA agarose (Protino, Macherey-Nagel, Germany). The purified CD3/GD2 BiTE protein was further characterized by SDS–PAGE and Western blot analysis.

## Reducing-PAGE and Western blotting

Protein analysis was conducted using SDS-polyacrylamide gel electrophoresis (SDS-PAGE) with a Mini-PROTEAN II system (Bio-Rad, CA, USA). Proteins were separated on 15% SDS-PAGE gels under denaturing conditions, and the bands were stained with InstantBlue™ (Sigma-Aldrich, USA). Additionally, separated protein bands were transferred onto a nitrocellulose membrane using a Bio-Rad Mini Trans-Blot cell. The nitrocellulose membrane was incubated in a blocking buffer (5% non-fat dry milk in TBST buffer: 10 mM Tris-HCl pH 8.0, 150 mM NaCl, and 0.05% Tween-20) for 1 h at room temperature. To detect human scFv CD3 protein, the membrane was incubated overnight at 4°C with mouse anti-E-Tag monoclonal antibody (GE Healthcare, UK). After 3 washes with TBST, AP-conjugated goat anti-mouse immunoglobulin (DAKO, Denmark) was added and incubated for 1 h at room temperature. Protein bands were visualized on the nitrocellulose membrane using the substrate (2,6-dichloroindophenol (DCIP)) (Sigma Chemical Co., USA). To detect CD3/GD3 BiTE protein, the membrane was incubated overnight at 4°C with anti-histidine tag antibody clone H8 (Sigma-Aldrich, USA). After 2 10-minute washing with TBST, the membrane was incubated for 2 h at room temperature with HRP-conjugated anti-mouse IgA + IgG + IgM (H+L) antibody (KPL, SeraCare, MA, USA). The specific protein bands were visualized using enhanced chemiluminescence (ECL) (Bio-Rad, CA, USA).

## Non-reducing PAGE

Non-reducing PAGE was performed to study the natural structure of the protein. Ten micrograms of CD3/GD2 BiTE protein were incubated with the reducing agent DTT (Affymetrix, Santa Clara, CA, USA) at various concentrations at 37°C for 30 minutes. The reduced protein was then subjected to 12% SDS-PAGE under native conditions and stained with InstantBlue™ (Sigma-Aldrich, USA).

## Bispecific antibodies binding assay

The binding activity of CD3/GD2 BiTE to target molecules, including CD3 and GD2 antigens, was evaluated by flow cytometry. GD2-positive cell line (SH-SY5Y), GD2-negative cell line (SK-N-SH), CD3-positive cell line (Jurkat), and CD3-negative cell line (Raji), at a cell concentration of $1 \times 10^5$ cells/ml, were pre-incubated with various concentrations of CD3/GD2 BiTE at 4°C for 1 h. After washing with 1 ml of PBS, cells were stained with 6xHis Tag antibody-Alexa 488 (Thermo Fisher Scientific, USA) and incubated at 4°C for 1 h. Cells were then washed twice and resuspended in 100 µl of PBS before analysis by flow cytometry (FACSCanto II, BD Biosciences, USA).

## *In vitro* cytotoxicity assay

SH-SY5Y cells were used as target tumor cells of the cytotoxicity assay. Briefly, $5 \times 10^4$ cells/ml of SH-SY5Y cells were stained with CellTrace™ CFSE (Thermo Fisher Scientific, USA) and added to 96-well flat-bottom plates (Costar,

Cambridge, USA) to incubate overnight. Next, activated Vγ9Vδ2 T cells were pre-incubated with CD3/GD2 BiTE for 30 min, then co-cultured with CFSE-stained SH-SY5Y cells at effector-to-target (E:T) ratios of 1:1, 1:5, and 1:10 for 24 h. After the incubation period, 7-AAD viability staining (Biolegend, United Kingdom) was performed, and SH-SY5Y cells stained with CFSE and 7-AAD (indicating the dead tumor cells) were analyzed by flow cytometry (FACSCanto II, BD Biosciences, USA). The percentage of dead tumor cells was calculated using the following formula:

$$\% \ Dead \ tumor \ cells \ = [\frac{dead \ tumor \ cells \ (CSFE+, \ 7-AAD+)}{total \ tumor \ cells \ (CSFE+, \ 7-AAD-)}] \times 100$$

## Three-dimensional (3D) tumor spheroid killing assay

To create 3D tumor spheroids, $2 \times 10^4$ cells/ml of SH-SY5Y GFP were cultured in medium with 2.5% Matrigel matrix (Corning Inc., NY, USA). Cells were plated into ultra-low attachment polystyrene plate (Corning Inc., NY, USA), centrifuged at $1000 \times g$ at 4°C for 10 min, and incubated for 72 h. Activated Vγ9Vδ2 T cells were stained with CellTrace™ Violet (Invitrogen, Thermo Fisher Scientific, USA) and incubated with CD3/GD2 BiTE before being added to the 3D tumor spheroid at an E:T ratio of 1:1. After 48 h of co-culture, ethidium homodimer-1 (Invitrogen, Thermo Fisher Scientific, USA) staining was performed to determine dead cells. The 3D tumor spheroid morphology and fluorescence intensity of GFP and ethidium homodimer were captured using Operetta CLS High-Content Analysis System (PerkinElmer, USA). The corrected total cell fluorescence (CTCF) of the 3D tumor spheroid was analyzed using ImageJ software and calculated using the following formula:

$$CTCF = Integrated \ Density \ - \ (Selected \ cell \ area \ \times MFI \ on \ background \ readings)$$

## Statistical analysis

The experiments were performed in triplicate, and the results are expressed as the means ± SD. For individual comparisons, statistical analysis was performed using one-way analysis of variance (ANOVA). Data with a statistical value of $P < 0.05$ were considered statistically significant.

## Results

### Discovery of human scFv CD3

We screened human scFv against CD3 epsilon from an M13 phage library displaying human scFv fragments using phage panning technology with recombinant CD3 epsilon. Following the elution of antigen-bound phages and their transformation into *E. coli* strain HB2151, ninety colonies that grew on LB agar containing ampicillin were randomly selected for human scFv gene insertion analysis by colony PCR. Sixty out of ninety colonies contained human scFv gene insertion. These sixty clones were cultured and induced for protein expression. Soluble human scFvs were extracted from the total cell lysate and their binding activity against CD3 epsilon was determined using indirect ELISA. Human scFv CD3 clones 18, 6, and 85 exhibited the highest signals in the indirect ELISA, respectively (Fig 1A). These three clones were selected for further expression and extraction of periplasmic and cytoplasmic proteins using fractionation buffer. Western blot analysis and indirect ELISA were performed on the fractionated proteins. As shown in Fig 1B, protein bands of human scFv CD3 clones 6, 18, and 85 were observed, with an approximate size of 28 kDa. No band were observed in fractionated proteins prepared from recombinant *E. coli* HB2151. Indirect ELISA results revealed that human scFv CD3 clone 18 markedly demonstrated the highest binding signal from both periplasmic and cytoplasmic fractions (Fig 1C). The amino acid sequence of human scFv CD3 clone 18, indicating VH-VL orientation and CDR regions, is shown in Fig 1D. Based

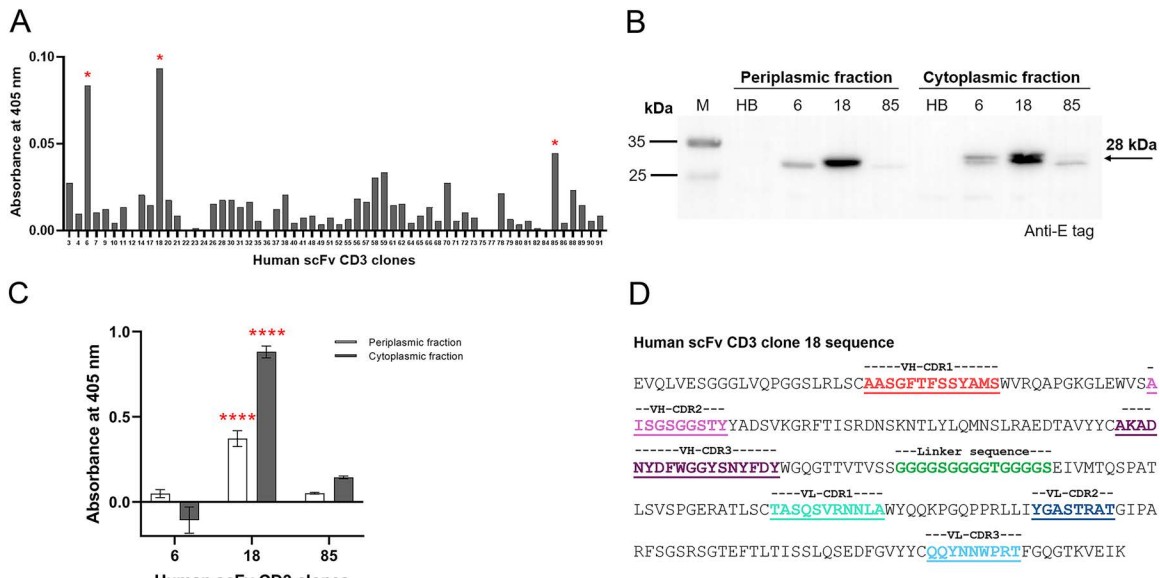

**Fig 1. Human scFv CD3 screening and characterization.** (A) Indirect ELISA screening of human scFv against CD3 epsilon. Sixty clones containing the human scFv gene insert were obtained from phage biopanning and induced for protein production. The total cell lysate of each clone was evaluated using indirect ELISA with CD3 epsilon as the antigen. (B) Periplasmic and cytoplasmic proteins of human scFv CD3 clones. Human scFv CD3 clones 6, 18, and 85 were expressed, and the cellular proteins were fractionated into periplasmic and cytoplasmic fractions. The fractionated proteins were analyzed by western blot using an anti-E tag antibody. (C) The binding activity of human scFv CD3 clones 6, 18, and 85 was evaluated using indirect ELISA. The white bar represents the signal from the periplasmic protein, and the gray bar represents the signal from the cytoplasmic protein. The absorbance of soluble human scFv CD3 against CD3 epsilon was analyzed by subtracting the absorbance of the negative protein in CD3 epsilon, followed by the absorbance of soluble human scFv CD3 in BSA-coated well, respectively. (D) The amino acid sequence of human scFv CD3 clone 18. The CDRs on the VH and VL regions are indicated by colored and underlined text. The flexible linker used to connect the VH and VL regions is highlighted in green.

on protein expression and the highest binding activity against CD3 epsilon, human scFv CD3 clone 18 was selected to construct a bispecific T cell engager targeting CD3 and GD2 molecules.

## CD3/GD2 BiTE construction and protein production

The mammalian cell expression vector, pCDNA3.1(+), containing the genes encoding CD3/GD2 BiTE (pCDNA3.1-CD3/GD2 BiTE), was constructed as illustrated in Figs 2A and 2B. The human scFv CD3 clone 18, orientated VH-VL, and the mouse scFv GD2, orientated VL-VH, were synthesized and linked by a 15-mer flexible linker, $(GGGGS)_3$. An interleukin-2 signal peptide was fused to the N-terminus to facilitate the secretion of the CD3/GD2 BiTE protein, while a polyhistidine tag was added to the C-terminus for detection and purification. To express the CD3/GD2 BiTE protein, HEK 293T cells were transfected with pcDNA3.1-CD3/GD2 BiTE and cultured for 72 h. After post-transfection, the cell culture medium containing the secreted CD3/GD3 BiTE protein was collected and purified using Ni-NTA agarose. Protein production and purity were confirmed by SDS-PAGE and Western blot analysis. As expected, a distinct single band of approximately 58 kDa was observed on reducing SDS-PAGE stained with InstantBlue™ (Fig 2C). Western blot analysis using an anti-His tag antibody confirmed the production and purification of the CD3/GD2 BiTE protein, showing the same 58 kDa band on the blot (Fig 2C). Additionally, protein dimerization was assessed by treating the CD3/GD2 BiTE protein with DTT before running non-reducing PAGE. As shown in Fig 2D, a single band was observed for the protein treated with 1 mM DTT, indicating that the CD3/GD2 BiTE protein remained monomeric. However, complete reduction occurred at 10 and 50 mM of DTT, resulting in a 58 kDa band on non-reducing PAGE. The production yield of CD3/GD2 BiTE was approximately

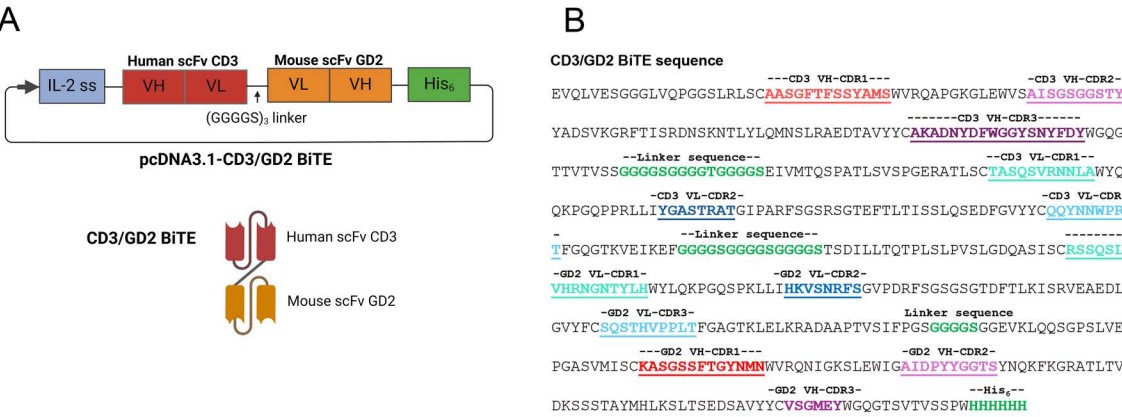

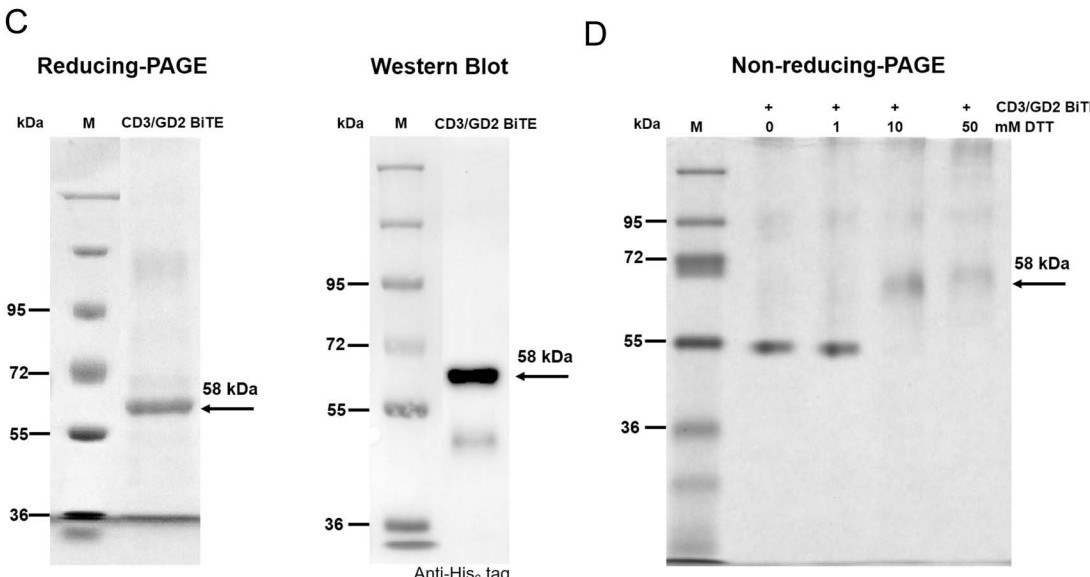

**Fig 2. Construction and expression of CD3/GD2 BiTE.** (A) Schematic diagram of CD3/GD2 BiTE in pcDNA3.1(+). CD3/GD2 BiTE consists of human scFv-CD3 clone 18 (VH-VL) linked to mouse scFv-GD2 (VL-VH) *via* a (G4S)3 linker. An Interleukin-2 signal peptide for protein secretion and a polyhistidine (His6) tag for protein purification and detection were added to the N-terminus and C-terminus of CD3/GD2 BiTE, respectively. A simplified structural model of CD3/GD2 BiTE is presented below. (B) Amino acid sequence of CD3/GD2 BiTE. The CDRs in the VH and VL regions of human scFv CD3 clone 18 and mouse scFv GD2 are indicated by colored and underlined text. The linkers and polyhistidine (His6) tag are highlighted in green. (C) Reducing-PAGE analysis of purified CD3/GD2 BiTE protein stained with InstantBlue and western blot analysis of purified CD3/GD2 BiTE detected using an anti-histidine tag antibody. The molecular weight of CD3/GD2 BiTEs is approximately 58 kDa. (D) Non-reducing-PAGE analysis of CD3/GD2 BiTE treated with DTT at different concentrations, stained with InstantBlue™. CD3/GD2 BiTE was dissociated by DTT and migrated on non-reducing-PAGE.

7.2 mg/L of culture medium, demonstrating that CD3/GD2 BiTE could be efficiently produced in mammalian cells with high yield and purity.

## Dual-binding activity of CD3/GD2 BiTE

To investigate the proper folding of CD3/GD2 BiTE, its binding activity to CD3 and GD2 antigens was evaluated using flow cytometry. CD3-positive cells (Jurkat cells), CD3-negative cells (Raji cells), GD2-positive cells (SH-SY5Y cells), and GD2-negative cells (SK-N-SH cells) were incubated with CD3/GD2 BiTE at various concentrations, followed by staining with an Alexa 488-conjugated 6xHis Tag antibody to measure the percentage of positive cells. When Jurkat cells were

incubated with CD3/GD2 BiTE at concentrations of 18, 180, and 900 nM, the percentage of positive cells was 15.03%, 57.63%, and 85.63%, respectively. The mean fluorescence intensity was elevated with higher concentrations of CD3/GD2 BiTE (Fig 3A). Similarly, the binding interaction of CD3/GD2 BiTE with SH-SY5Y cells appeared a comparable pattern to that observed with Jurkat cells (Fig 3B). In contrast, the percentage of positive cells for Raji and SK-N-SH cells remained below 10% at all tested concentrations of CD3/GD2 BiTE. In conclusion, CD3/GD2 BiTE demonstrated strong and specific binding to its targets, the CD3 molecule on Jurkat cells and the GD2 molecule on SH-SY5Y cells, in a dose-dependent manner.

### *In vitro* cytotoxicity enhancement of CD3/GD2 BiTE and activated Vγ9Vδ2 T cells

Initially, Vγ9Vδ2 T cell populations were expanded *in vitro* and characterized using flow cytometry. As shown in Fig 4A, the immunophenotyping characterization indicated that $3.39 \pm 2.67\%$ of γδT cells were present in the initial PBMCs at a concentration of $1 \times 10^6$ cells/ml on day 0. After 14 days of expansion culture, the Vγ9Vδ2 T cell phenotype increased to $88.56 \pm 10.32\%$, with a fold expansion of $103.29 \pm 50.76$. The purity of the expanded Vγ9Vδ2 T cells was $82.32 \pm 8.92\%$.

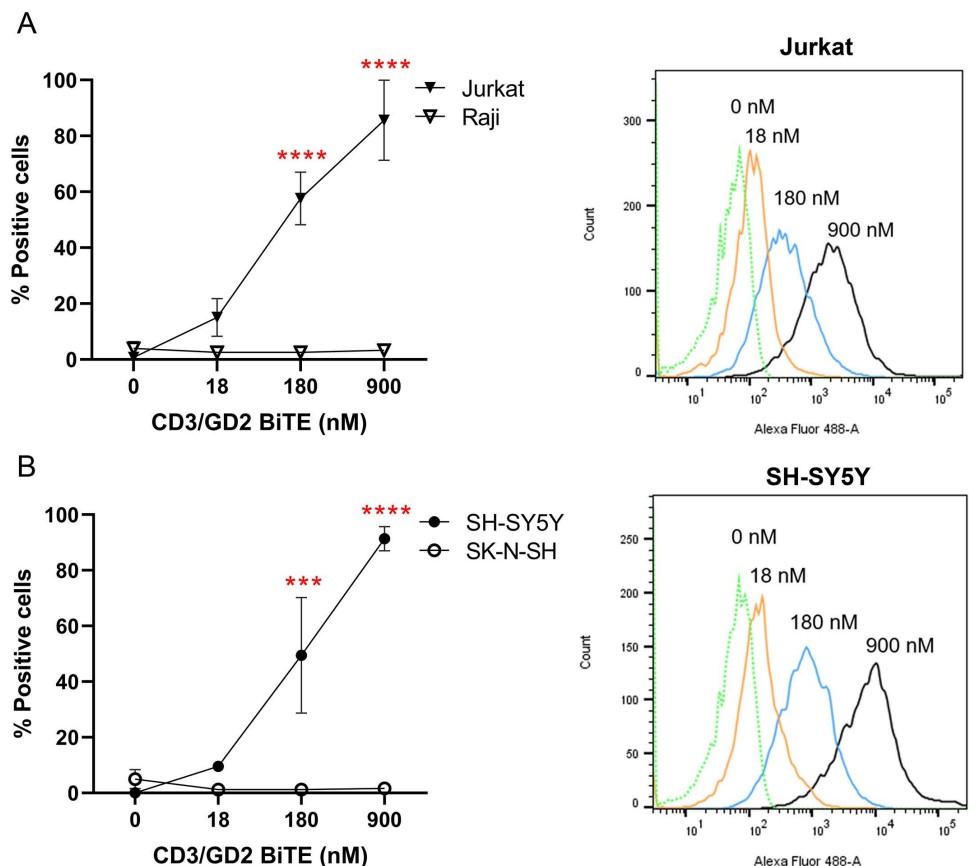

**Fig 3. Binding activity of CD3/GD2 BiTE on its target cells.** (A) CD3-positive Jurkat cells and CD3-negative Raji cells (B) GD2-positive SH-SY5Y cells and GD2-negative SK-N-SH cells. Jurkat, Raji, SH-SY5Y, and SK-N-SH cells were incubated with various concentrations of CD3/GD2 BiTE, and binding was determined by flow cytometry using Alexa Flour 488-conjugated 6xHis Tag antibody. The mean percentage of positive cells is presented in the line graph. A mock protein was used as non-specific control. The mean fluorescence intensity (MFI) histograms of Jurkat and SH-SY5Y cells are shown on the right. The experiments were performed in triplicate, and individual results are plotted, with the means±SD presented. *$p < 0.05$, **$p < 0.01$, ***$p < 0.001$.

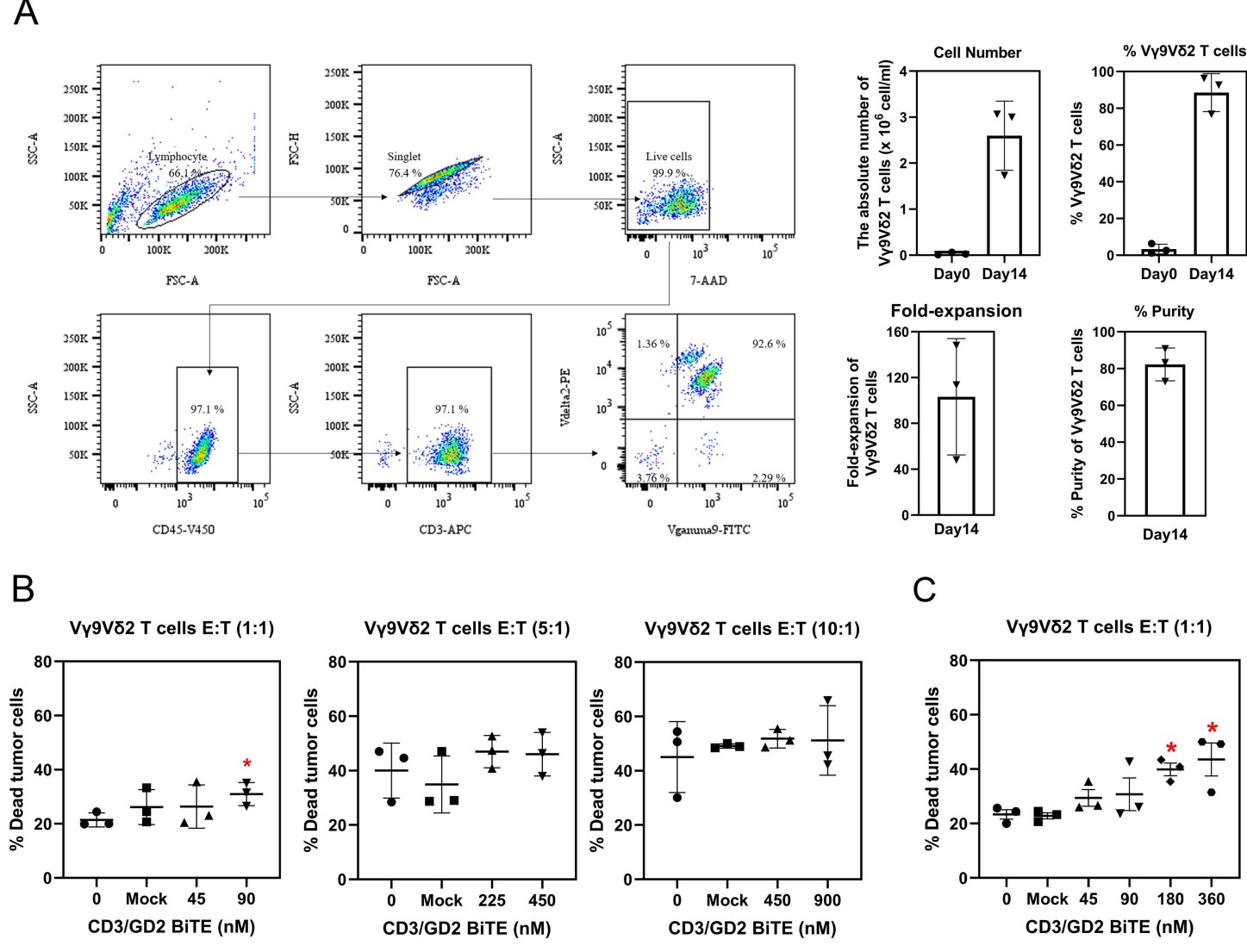

**Fig 4. *In vitro* cytotoxicity of CD3/GD2 BiTE and Vγ9Vδ2 T cells against SH-SY5Y cells at various E:T ratios.** (A) Immunophenotyping characterization of activated Vγ9Vδ2 T cells by flow cytometry. The cell number and percentage of Vγ9Vδ2 T cells on Day0 and Day14 are shown in the bar graph. The fold-expansion and purity on Day14 are also presented in the bar graph. (B) The mean percentage of dead tumor cells is plotted against the concentration of CD3/GD2 BiTE (nM) at E:T ratios of 1:1, 5:1, and 10:1, respectively. (C) The mean percentage of dead tumor cells from Vγ9Vδ2 T cells primed with various concentrations of CD3/GD2 BiTEs at an E:T ratio of 1:1. CD3/GD2 BiTE was incubated with activated Vγ9Vδ2 T cells prior to being added to CFSE-stained SH-SY5Y cells for 24 h. Dead cells were stained with 7-AAD. Dead tumor cells presenting both CFSE and 7-AAD were indicated and quantified by flow cytometer. The dot plots were calculated from three independent experiments, and the results are expressed as means ± SD. *$p < 0.05$.

These findings confirmed that Vγ9Vδ2 T cells were activated and proliferated effectively using ZOL and IL-2, reaching sufficient numbers for subsequent *in vitro* cytotoxicity experiments.

To evaluate whether the CD3/GD2 BiTE enhances the cytotoxicity of Vγ9Vδ2 T cells *in vitro*, SH-SY5Y neuroblastoma cells were used as target cells, while activated Vγ9Vδ2 T cells served as effector cells at different effector-to-target (E:T) ratios of 1:1, 5:1, and 10:1. The CD3/GD2 BiTE concentrations that correspond to 50% and 100% binding activity were utilized to perform the cytotoxicity assay. As shown in Fig 4B, the mean percentage of dead tumor cells at E:T ratios of 1:1, 5:1, and 10:1 with Vγ9Vδ2 T cells alone was 21.43±2.57%, 40.00±10.12%, and 45.03±13.07%, respectively. The

combination of CD3/GD2 BiTE with activated Vγ9Vδ2 T cells increased the percentage of dead tumor cells at all E:T ratios, with significant enhancement in cytotoxicity effect observed only at E:T ratio of 1:1. Specifically, the percentage increased from 21.43±2.57% without CD3/GD2 BiTE to 30.97±4.27% ($p < 0.005$) with 90 nM CD3/GD2 BiTE (Fig 4B). At E:T ratios of 5:1 and 10:1, the combination did not significantly improve cytotoxicity against SH-SY5Y cells (Fig 4B). Therefore, the combination of CD3/GD2 BiTE and activated Vγ9Vδ2 T cells at E:T ratio of 1:1 was selected to perform further cytotoxicity assay.

To further investigate the cytotoxic function of activated Vγ9Vδ2 T cells mediated by CD3/GD2 BiTE, concentrations of 45, 90, 180, and 360 nM, corresponding to 50%, 100%, 200%, and 400% binding activity to effector cells, were used. As shown in Fig 4C, tumor cell death increased in dose-dependent manner. Significant cytotoxic effects were observed at CD3/GD2 BiTE concentrations of 180 and 360 nM, with 39.90±4.09% and 43.57±10.46% dead tumor cells, respectively. This indicated that the combination of CD3/GD2 BiTE enhanced the cytotoxicity 1.70-fold and 1.86-fold higher compared to Vγ9Vδ2 T cell alone. The mock control showed no significant differences across all E:T ratios, without CD3/GD2 BiTE treatment. Notably, cytotoxicity of CD3/GD2 BiTE at 360 nM with an E:T ratio of 1:1 (Fig 4C) was comparable to that of activated Vγ9Vδ2 T cells alone at E:T ratios of 5:1 and 10:1 (Fig 4B), demonstrating that CD3/GD2 BiTE enhanced tumor cell killing with 5-10 times fewer effector cells.

### Anti-tumor killing activity of CD3/GD2 BiTE and Vγ9Vδ2 T cells on 3D tumor spheroid

The killing ability of the combination of CD3/GD2 BiTE and Vγ9Vδ2 T cells was evaluated using a 3D *in vitro* model, which mimics the microenvironment and complexity of native neuroblastoma tumors. The 3D tumor spheroids were generated from SH-SY5Y GFP cells. After a 48 h co-culture of Vγ9Vδ2 T cells armed with CD3/GD2 BiTE and the 3D tumor spheroids at an E:T ratio of 1:1, the fold change of the concentrated total cell fluorescence (CTCF) of live SH-SY5Y GFP cells and dead cells stained with ethidium homodimer-1 was calculated in comparison to the outcomes of the cell systems without BiTE addition. As shown in Fig 5, the GFP signal from the 3D tumor spheroids slightly decreased when treated with Vγ9Vδ2 T cells armed with CD3/GD2 BiTE. The reduction in the number of live target cells was significant, at 360 nM of CD3/GD2 BiTE ($p < 0.05$). Meanwhile, the EthD-1 signal, indicating dead cells, significantly increased in a dose-dependent manner, ranging from a 1.75 to 2.20-fold CTCF change. Notably, 180 nM of CD3/GD2 BiTE exhibited the highest CTCF fold change at 2.20-fold ($p < 0.01$) (Fig 5B). The mock control showed no significant difference in CTCF fold change for both live and dead cells. These results indicate that the combination of CD3/GD2 BiTE and Vγ9Vδ2 T cells effectively demonstrated killing capability against neuroblastoma in the 3D tumor spheroid model.

### Discussion

Immunotherapy has emerged as a promising strategy for treating a broad range of malignant tumors. Anti-GD2 monoclonal antibodies have shown potent anti-tumor activity against neuroblastoma and are already incorporated into conventional treatment protocols. These therapeutic antibodies induce neuroblastoma cytotoxicity *via* Fc-dependent mechanisms, including complement-dependent cytotoxicity (CDC), antibody-dependent cellular cytotoxicity (ADCC) mediated by NK cells, and antibody-dependent cellular phagocytosis (ADCP) [33]. The first generation of anti-GD2 antibody (3F8) originated from a murine source and was later replaced by a chimeric antibody to reduce human anti-mouse antibody (HAMA) responses [34]. Over the past decade, the concept of a bispecific antibody targeting CD3 on T cells and GD2 on neuroblastoma cells has been developed. Bispecific antibodies function by linking tumor cells and T cells, activating the latter to release cytotoxic cytokines for targeted tumor cell lysis [35,36]. In this study, we isolated a new human scFv specific for CD3 epsilon from a naïve human scFv gene library using phage panning technology (Fig 1A). The human scFv CD3 clone 18, which demonstrated protein expression (Fig 1B) and the highest binding activity against CD3 in indirect ELISA (Fig 1C), was selected to construct the chimeric bispecific antibody, called CD3/GD3 BiTE.

CD3/GD2 BiTE is a recombinant, Fc-less, non-immunoglobulin G (IgG)-like antibody constructed by linking two single-chain antibody fragment sequences with a flexible linker. Human scFv CD3 clone 18 is fused to mouse scFv GD2 using a 15-mer flexible linker (GGGGS)$_3$, forming a bispecific, bivalent scFv antibody molecule (Figs 2A and 2B). The protein, CD3/GD2 BiTE was successfully expressed and purified in a mammalian expression system, yielding approximately 7.2 mg/L of culture medium (Fig 2C). HEK 293T cells are one of the most common mammalian cell lines employed for the production of therapeutic biopharmaceuticals both by transient transfection and stable cell line expression systems [37,38]. Single chain antibodies naturally fold from a primary amino acid sequence to a three-dimensional structure stabilized by interchain disulfide bonds between VH and VL domains. Under specific conditions, scFvs may form dimers, depending on the linker length, antibody sequence, and other external factors [39]. To investigate whether CD3/GD2 BiTE formed dimers, we tested the protein under reducing conditions using DTT as a reducing agent. CD3/GD2 BiTE, both with and without DTT, adopted a compact shape, resulting in faster protein migration compared to the denatured form (Fig 2D). Notably, only a single band was observed at the same size for both CD3/GD2 BiTE without DTT and the DTT-treated (1mM) on non-reducing gel (Fig 2D), indicating the absence of protein dimerization. Thus, CD3/GD2 BiTE was confirmed to be a monomeric protein. The use of mammalian expression system enabled proper folding and post-translation modifications, resulting in high productivity and biological activity [40]. However, large-scale recombinant protein production, which enhances both yield and quality, remains essential to accelerate the manufacturing of these therapeutic antibodies [41].

The CD3/GD2 BiTE was constructed as a single-chain protein comprising 521 amino acids. This bispecific antibody features three peptide linkers: one between the VH and VL domains of human scFv CD3 clone 18, another between the VL and VH domains of mouse scFv GD2, and a third linking human scFv CD3 clone 18 to mouse scFv GD2 (Fig 2B). Additionally, each scFv antibody contains two interchain disulfide bonds between the VH and VL domains. The intricate structure of CD3/GD2 BiTE may influence proper protein folding, potentially reducing yield and binding activity [42]. However, the protein displayed a compact size on the non-reducing gel (Fig 2D) and retained strong antigen recognition abilities, specifically binding CD3 and GD2 on Jurkat and SH-SY5Y cells, respectively (Figs 3A and 3B). These findings confirm that CD3/GD2 BiTE was correctly produced and well-folded when transiently expressed in HEK 293T cells.

Traditionally, CD8+ T cells are considered the optimal effector cells for cancer immunotherapy. However, BiTEs have the ability to activate various CD3+ effector cells in PBMCs, including CD3+CD8+ cells, CD3+CD4+ cells, and CD3+ NK cells [43]. Most bispecific antibodies targeting GD2 and CD3 have been shown to activate T cells, promoting cytotoxicity against neuroblastoma. For instance, T cells armed with 3F8BiAb have demonstrated specific cytotoxicity against GD2-positive neuroblastoma cell lines [44]. Similarly, T cells equipped with anti-GD2 bispecific antibodies (anti-GD2 BsAb) have exhibited significant anti-tumor activity against osteosarcoma both *in vitro* and *in vivo* [45]. Therefore, in this study, we investigated the functionality of CD3/GD2 BiTE regarding T cell activation and redirection of neuroblastoma cell lysis. We conducted an *in vitro* cytotoxicity assay using PBMCs as effector cells. PBMCs were primed with CD3/GD2 BiTE at concentrations of 18 and 180 nM before being added to CSFE-stained SH-SY5Y cells at E:T ratios of 1:1 and 10:1 for 24 h. The percentage of dead tumor cells in PBMCs armed with CD3/GD2 BiTE did not significantly differ from PBMCs without CD3/GD2 BiTE or the mock control (S1A Fig). To further evaluate BiTE-mediated T cell activation, PBMCs were co-cultured with CD3/GD2 BiTE at concentrations of 18 and 360 nM for 24 h. Subsequently, PBMCs were stained with antibodies specific to T cell activation markers CD25 and CD69. The percentage of CD25+ and CD69+ cells in PBMCs co-cultured with CD3/GD2 BiTE was less than 10%, which was comparable to PBMCs without BiTE treatment (S1B Fig). These results indicate that CD3/GD2 BiTE only recognized the CD3 molecule but did not activate T cells or mediate cytotoxicity against neuroblastoma cell lines. Thus, PBMCs were not suitable effector cells for CD3/GD2 BiTE.

The adoptive transfer of expanded Vγ9Vδ2 T cells has been shown to be safe and well-tolerated in clinical trials. γδ T cell-based immunotherapy, especially Vγ9Vδ2 T cells, holds significant promise for tumor treatment due to low toxicity

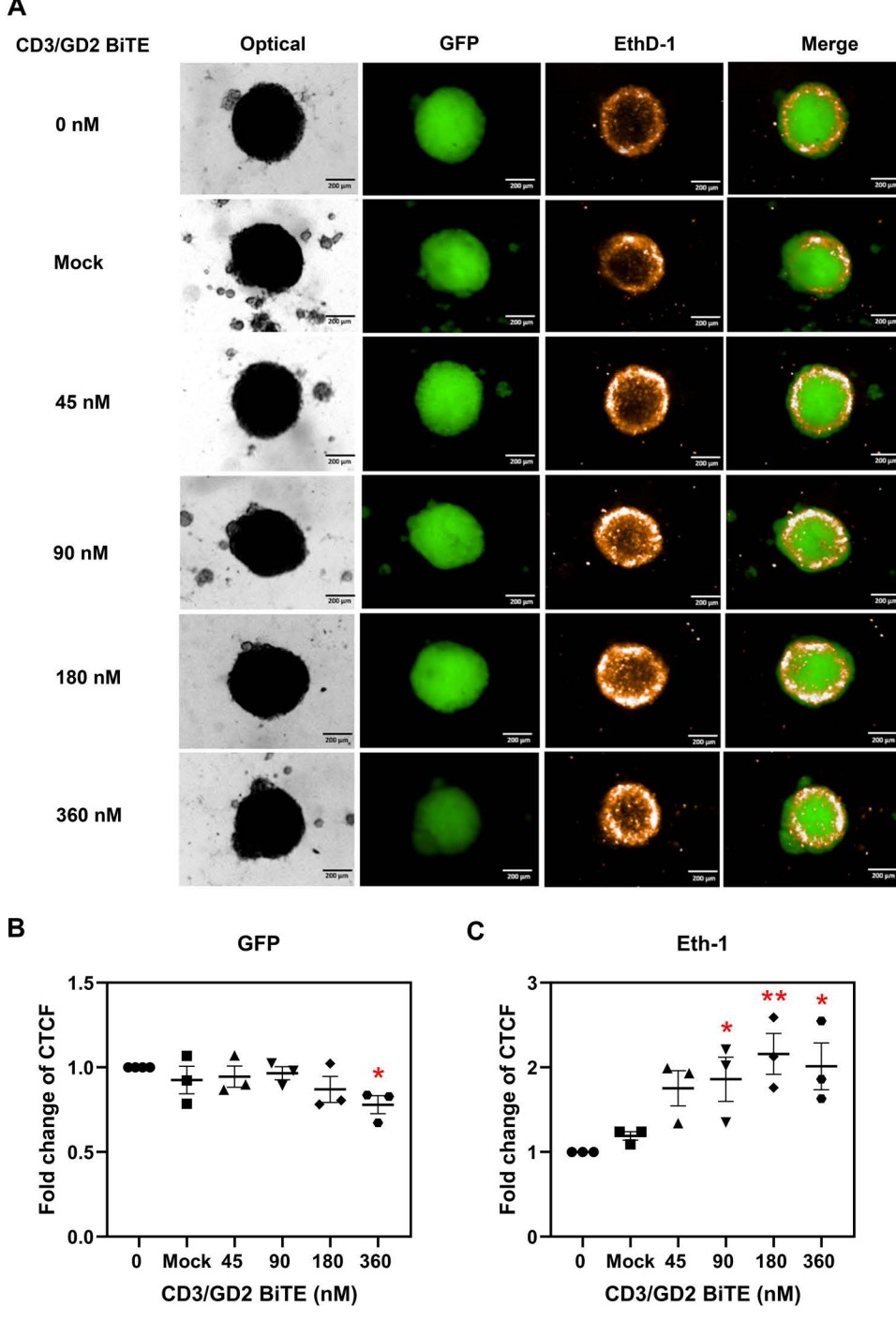

**Fig 5. The 3D tumor spheroid killing effect of combined CD3/GD2 BiTE and Vγ9Vδ2 T cells.** (A) Fluorescence images of 3D tumor spheroids treated with CD3/GD2 BiTE-armed Vγ9Vδ2 T cells. (B) The CTCF fold change of GFP and EthD-1 in 3D tumor spheroids treated with CD3/GD2 BiTE-armed Vγ9Vδ2 T cells. The 3D tumor spheroids of neuroblastoma were generated using SH-SY5Y GFP cells. CD3/GD2 BiTE-armed Vγ9Vδ2 T cells were prepared and added to the 3D tumor spheroids at an E:T ratio of 1:1 for 48 h. Live cells expressed GFP, and dead cells were stained with EthD-1. The fold change of CTCF for GFP and EthD-1 was calculated relative to non-treated CD3/GD2 BiTE. Three independent experiments were performed. The results are plotted individually, and the means ± SD are presented. *$p < 0.05$, **$p < 0.01$.

profile and lack of dependency on an individual's major histocompatibility complex, making therapeutic γδ T cells from healthy donors a viable option to reduce on-target off-tumor effects [46]. Activated γδ T cells inherently have the ability to kill a broad spectrum of solid tumors and hematological malignancies, including neuroblastoma [20,47–49]. Strategies to enhance the tumor-targeting ability of activated γδ T cells using bispecific antibodies have been explored although, the combination of bispecific antibodies and Vγ9Vδ2 T cells for neuroblastoma has yet to be reported in clinical study or clinical trials [50]. In our study, we report for the first time that CD3/GD2 BiTE combined with activated Vγ9Vδ2 T cells significantly enhances the killing of neuroblastoma cells. At an E:T ratio of 1:1, CD3/GD2 BiTE at concentrations of 180 and 360 nM increased cytotoxicity of activated Vγ9Vδ2 T cells to 38.10±4.20% and 42.30±4.45%, respectively, (Figs 4B and C) Notably, these concentrations achieved cytotoxicity effects equivalent to those seen with effector cells alone at E:T ratios of 5:1 and 10:1, respectively, indicating that CD3/GD2 BiTE reduces the required number of Vγ9Vδ2 T cells by five- to ten-fold. Typically, high E:T ratios are needed to achieve effective tumor cell killing, which is a challenge since Vγ9Vδ2 T cells constitute only about 0.5-5%. of peripheral blood lymphocytes. Large-scale expansion of Vγ9Vδ2 T cells has been difficult due to variability in *ex vivo* culture protocols, including differences in culturing conditions, timing, and the use of zoledronic acid and IL-2 co-stimulators, which can result in diverse phenotypes and effector characteristics [51]. The combination of bispecific antibodies, such as CD3/GD2 BiTE, with Vγ9Vδ2 T cells represents a promising strategy to improve tumor targeting and cytotoxic using fewer effector cells. CD3/GD2 BiTE effectively redirected Vγ9Vδ2 T cells to target neuroblastoma cells in both monolayer cultures and 3D spheroid models (Fig 5). The 3D spheroids, which mimic the microenvironment, characteristics, and morphology of solid tumors, provided a more representative model compared to 2D cultures [52]. Although our study demonstrated enhanced cytotoxicity in CD3/GD2 BiTE combined with Vγ9Vδ2 T cells on 2D and 3D cell cultures, the efficacy and safety of this therapeutic combination need to be further validated in *in vivo* models and clinical trials. Allogenic γδ T cells provide several advantages in cancer immunotherapy which are MHC-independent tumor recognition and reduced risk of graft-versus-host disease (GVHD [53,54]. In addition, the mass production of "off-the-shelf" allogeneic γδ T cells can be produced since γδ T cells can be expanded, manufactured, and stored which are readily available for use in multiple patients [55,56]. Therefore, allogeneic γδ T cells derived from a heathy donor could be considered to combine with CD3/GD2 BiTE for further verification the anti-tumor efficacy in animal models and clinical trials.

In conclusion, we successfully isolated a human scFv against CD3 epsilon using phage panning display technology. The selected human scFv CD3 clone 18, which exhibited the highest binding activity in indirect ELISA, was used to construct a chimeric bispecific antibody, in the format of CD3/GD2 BiTE. This bispecific molecule, comprising human scFv CD3 clone 18 and mouse chimeric bispecific antibody, scFv GD2 connected via a flexible linker, was well-expressed and properly folded as a monomeric protein in a mammalian expression system. CD3/GD2 BiTE retained high specificity for its target antigens, CD3 and GD2, on Jurkat and SH-SY5Y cells, respectively. The cytotoxic mechanism of combined CD3/GD2 BiTE and activate Vγ9Vδ2 T cells against neuroblastoma cells was illustrated in Fig 6. Initially, activated Vγ9Vδ2 T cells are primed with CD3/GD2 BiTE. The scFv CD3 domain binds to CD3 molecule on Vγ9Vδ2 T cells whereas scFv-GD2 domain is free to bind GD2 molecule on neuroblastoma cells. CD3/GD2 BiTE-primed activated Vγ9Vδ2 T cells which has been improved the binding specificity and avidity against GD2 molecule are capable of targeting neuroblastoma cells. Cytotoxic granules from activated Vγ9Vδ2 T cells are released and induced apoptosis in neuroblastoma cells. Notably, the CD3/GD2 BiTE significantly enhanced tumor cell targeting and anti-tumor activity of activated Vγ9Vδ2 T cells in both monolayer cell cultures and 3D spheroid models *in vitro*. Our study proposes a novel approach for neuroblastoma immunotherapy using a combination of BiTE and human Vγ9Vδ2 T cells which could offer promising benefits for neuroblastoma patients in the future.

## Supporting information

**S1 Fig.** *In vitro* **cytotoxicity of CD3/GD2 BiTE and PBMC on neuroblastoma cell line and PBMC activation by CD3/GD2 BiTE.** (A) The mean percentage of dead tumor cells from PBMC armed with CD3/GD2 BiTE at E:T ratios of 1:1 and 10:1. Human PBMCs were primed with CD3/GD2 BiTE at concentrations of 18 and 180 nM before being

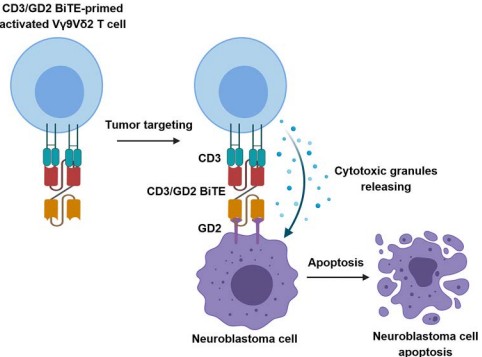

**Fig 6. Schematic illustration of the cytotoxic mechanism of combined CD3/GD2 BiTE and activate Vγ9Vδ2 T cells against neuroblastoma cells.** Activated Vγ9Vδ2 T cells are primed with CD3/GD2 BiTE. The scFv CD3 domain binds to CD3 molecule on Vγ9Vδ2 T cells whereas scFv-GD2 domain is free to bind GD2 molecule on neuroblastoma cells. CD3/GD2 BiTE-primed activated Vγ9Vδ2 T cells which has been improved the binding specificity and avidity against GD2 molecule are capable of targeting neuroblastoma cells. Cytotoxic granules from activated Vγ9Vδ2 T cells are released and induced apoptosis in neuroblastoma cells.

added to CFSE-stained SH-SY5Y cells for 24 h. Dead cells were stained with 7-AAD. Dead tumor cells positive for both CFSE and 7-AAD were detected and measured using flow cytometer. (B) The expression of activation markers CD25 and CD69 on PBMCs after co-culture with CD3/GD2 BiTE. The PBMCs were co-cultured with CD3/GD2 BiTE at concentrations of 18 and 360 nM for 24 h. PBMCs were then stained with anti-CD25 or anti-CD69 antibodies and analyzed by flow cytometry. The dot plots were generated from three independent experiments, and the results are expressed as means ± SD.
(TIF)

**S1 File. CD3/GD2 BiTE Raw data.**
(ZIP)

## Acknowledgments

We would like to extend our appreciation for the Faculty of Medical Technology and the Department of Pediatrics, Faculty of Medicine Ramathibodi Hospital, Mahidol University for supporting laboratory facilities.

## Author contributions

**Conceptualization:** Kuntida Kitidee, Suradej Hongeng.

**Data curation:** Kuntida Kitidee.

**Formal analysis:** Kuntida Kitidee, Sumet Amonyingcharoen.

**Funding acquisition:** Usanarat Anurathapan, Suradej Hongeng.

**Investigation:** Kuntida Kitidee, Sumet Amonyingcharoen, Sarinthip Preedagasamzin, Korakot Atjanasuppat, Piamsiri Sawaisorn, Pornprapa Srimorkun.

**Methodology:** Kuntida Kitidee, Sumet Amonyingcharoen, Sarinthip Preedagasamzin, Korakot Atjanasuppat, Piamsiri Sawaisorn, Pornprapa Srimorkun, Sawang Petvises, Wanpen Chaicumpa, Suparerk Borwornpinyo, Usanarat Anurathapan, Suradej Hongeng.

**Project administration:** Kuntida Kitidee, Sumet Amonyingcharoen, Suradej Hongeng.

**Resources:** Sawang Petvises, Wanpen Chaicumpa, Suparerk Borwornpinyo, Usanarat Anurathapan, Suradej Hongeng.

**Supervision:** Kuntida Kitidee, Suradej Hongeng.

**Validation:** Kuntida Kitidee, Sumet Amonyingcharoen, Usanarat Anurathapan, Suradej Hongeng.

**Visualization:** Kuntida Kitidee, Sumet Amonyingcharoen, Suradej Hongeng.

**Writing – original draft:** Kuntida Kitidee, Sumet Amonyingcharoen.

**Writing – review & editing:** Kuntida Kitidee, Suradej Hongeng.

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
