## [Decision Letter · Decision Letter 0]

9 Aug 2024

PONE-D-24-27594Combining CD3/GD2 bispecific T cell engagers with human Vγ9Vδ2 T cells facilitates neuroblastoma cell killingPLOS ONE

Dear Dr. hongeng,

Thank you for submitting your manuscript to PLOS ONE. After careful consideration, we feel that it has merit but does not fully meet PLOS ONE’s publication criteria as it currently stands. Therefore, we invite you to submit a revised version of the manuscript that addresses the points raised during the review process.

We look forward to receiving your revised manuscript.

Kind regards,

Mahmood S Choudhery, PhD

Academic Editor

PLOS ONE

Journal Requirements:

4. Thank you for stating the following financial disclosure:"The Research Chair Grant from the National Science and Technology Development Agency, Thailand (FDA-CO-2559-3325-TH)

The Fundamental Fund from Thailand Science Research and Innovation (FRB650007/0185) "  

5. Thank you for stating the following in the Acknowledgments Section of your manuscript: "This work was supported by the Research Chair Grant from the National Science and Technology Development Agency, Thailand (FDA-CO-2559-3325-TH) to Suradej Hongeng and the Fundamental Fund from Thailand Science Research and Innovation (FRB650007/0185) to Usanarat Anurathapan. The studies involving human participants were reviewed and approved by the ethical clearance committee on human rights to research involving human subjects, Faculty of medicine Ramathibodi hospital, Mahidol university, No. (MURA2020/879). The funders had no role in study design, data collection and analysis, decision to publish, or preparation of the manuscript."

Please remove any funding-related text from the manuscript and let us know how you would like to update your Funding Statement. Currently, your Funding Statement reads as follows: "The Research Chair Grant from the National Science and Technology Development Agency, Thailand (FDA-CO-2559-3325-TH)

The Fundamental Fund from Thailand Science Research and Innovation (FRB650007/0185) "

Additional Editor Comments:

The study’s rationale could be more clearly written. The methodology section requires additional specific information to ensure reproducibility. The current version of Figure 2 is blurred, which compromises the clarity of the data presented. Please provide a revised version with higher resolution and clarity, ensuring that all labels and data points are clearly visible. Some of the single-panel figures could be combined for a more streamlined presentation.

Reviewers' comments:

Reviewer's Responses to Questions

**Comments to the Author**

1. Is the manuscript technically sound, and do the data support the conclusions?

Reviewer #1: Partly

Reviewer #2: Yes

Reviewer #3: Partly

2. Has the statistical analysis been performed appropriately and rigorously? 

Reviewer #1: Yes

Reviewer #2: No

Reviewer #3: Yes

3. Have the authors made all data underlying the findings in their manuscript fully available?

Reviewer #1: No

Reviewer #2: Yes

Reviewer #3: Yes

4. Is the manuscript presented in an intelligible fashion and written in standard English?

Reviewer #1: No

Reviewer #2: No

Reviewer #3: Yes

5. Review Comments to the Author

Reviewer #1: In the present manuscript, the authors present the construction of bispecific T-cell engager composed of an anti-GD2 and an anti-CD3 targeting unit, to activate the gamma/delta T cells for killing in response to a CD2-positive neuroblastoma cell line. They expressed the molecule at a high concentration in HEK cells, purified it with a Ni-NTA-based affinity chromatography, and tested it for the activity. The latter included binding to both types of antigen-positive cell lines (effector and target), and the killing activity for the target cells. The BiTE activity is also ex.amined in a 3D culture of tumor cells (as a matrix-embedded cell line), where armed gamma/delta T cells also show killing capability as judged by fluorescent microscopy.

The concept of the study is very valuable, as (1) the effector cell group used here, the gamma/delta T cells, emerged in the last decades as a powerful mechanism of tumor combat, and (2) the neuroblastoma is a condition where novel therapy concepts are certainly required. However, there is a number of questions to be addressed:

- Discovery of anti-CD3 molecule: all data on discovery and characterization are missing, and how was it made sure that the chosen clone is not an agonist to CD3? Outcomes to procedures described in lines 155-159: the results of anti-CD3 epsilon screening with sequencing of relevant clones, and the results of the activity of the selected CD3 clones (basic biophysical characterization with SDS-PAGE, activity with ELISA, including potential cross-reactivity with unrelated antigens, and sequencing for identity). How was the decision on using this particular anti-CD3 antibody based, and in particular, how was the agonism excluded?

- Characterization of the chosen BiTE molecule: for validation of binding data, a size-exclusion chromatography profile showing that the studied molecule is a monomer, must be shown.

- Conceptual: why is this BiTE expected to engage with gamma/delta T cells, if CD3 is present also on several other T cells? How does this BiTE differentiate form other BiTEs in this aspect? A control anti-CD3 molecule would be very valuable for all experiments.

- Methodology: it must be shown that the anti-CD3 clone is specific to CD3 (by using a CD3-negative T-cell line for experimental control), and that the BiTE is not active on GD2-negative cells.

- Results and outlook: the molecule (if negative controls which are to be delivered indeed confirm the data presented) is not very potent as over 100 nM concentrations are required for the effects. Is this due to low affinity towards any or both of the antigens? Which measures will be taken to achieve a more potent activation of T cells?

- Lines 136-139: the results of monitoring of gamma/delta T cells characterization and activation?

Please find below a list of remarks which I hope will be helpful.

Lines 8-9: “We created a bispecific T cell engagers antibody against CD3 and disialoganglioside (GD2), called CD3/GD2 BiTEs“ – a very particular use of singular and plural, please correct

Line 60: “is currently being revolutionized“ – remodelled would be a better expression; one part of the molecule is kept constant while the TAA-targeting part is changed for another specificity

Line 63: CD3 molecule

Lines 64-66: this sentence is not ok and needs to be re-written

Line 71: “GD2 can be considered as a tumor associated antigen which is an important target for the treatment of NB“: please reword: GD2 can be considered as a tumor associated antigen and important target for the treatment of NB

Line 78: “Despite, monoclonal anti-GD2 antibody has potentially suitable” – Despite the potential suitability of the monoclonal anti-GD2 antibody

Line 84: “The predominant type of γδ cells in human peripheral blood“

Line 90: “the safety of Vγ9Vδ2 T cells-based immunotherapy in hematologic malignancies has been reported“ – the safety issues with … have been reported

Line 93: “for more than three decades” – since more than three decades

Line 101: bispecific T-cell engager antibody, and please correct throughout the text

Line 106: “on neuroblastoma cell line and 3-dimentional culture” – and in 3-dimensional culture

Line 115: “T cell leukemia cell lines, Jurkat were” – T-cell leukemia cell lines and Jurkat cells

Lines 118-119: “were incubated at 37˚C, 5% CO2 and 95% humidified atmosphere“: CO2, 2 in subscript, and please check the percentage of humidification (usually it is around 80)

Line 132: “RPMI 11640 medium“ – 1640

Line 137: “cells were characterized the cell surface phenotype” – for the cell surface phenotype

Line 138, and throughout the manuscript: antibodies should be cited with RRIDs

Line 150: how many wells were used for panning? And “Unbound phages were removed by the washing buffer 10 times” – volume of washing buffer

Line 151: “E. coli strain HB2151 was added into the well” – the volume of the exponential culture per well and in total?

Line 156: “tested the binding activity of soluble human” – tested for the binding activity

Line 157:” HRP conjugated anti-E tag” – please specify the reagent

Line 178: “CD3/GD2 BiTEs protein at 2-10 μg was performed SDS-polyacrylamide gel

electrophoresis“ – surely, the analysis of the protein was performed

Line 180: “On the other hand,” – additionally is a better expression

Line 259: “measure the number of CD3/GD2 BiTEs bound on the target cells“ – the number is not really determined here, so the binding activity or similar would be a better expression

Line 264: was elevated

Line 265: “exhibited similar characterizations” – appeared similar

Line 267: “Consequently, the CD3/GD2 BiTEs demonstrated” – To conclude, the CD3/GD2…

Line 297: nM (nanomolar), where the percentage

Line 299: word order: “enhanced 1.57-fold and 1.75-fold higher cytotoxicity compared” – enhanced the cytotoxicity 1.57-fold and 1.75-fold

Line 300: “Mock control demonstrated the similar dead tumor cell percentage of non-treated CD3/GD2 BiTEs“ – this is an important statement and should not be affected by poor grammatic structure – in mock control, the percentage of dead tumor cells was similar to when the cells were not treated with CD3/GD2 BiTEs- does it hold true for all E:T ratios?

Lines 304-306: important result, again not easy to understand: “using CD3/GD3 BiTEs to increase the specificity of activated Vγ9Vδ2 T cells against neuroblastoma cells reduced five to tentimes in the number of effector cells” – the addition of the CD3/GD2 BiTEs enabled tumor cell killing at 5-10-lower number of effector cells.

Line 315: “with non-treated CD3/GD2 BiTEs” – with the outcomes of the cell systems without BiTe addition.

Lines 317-318: “The CTCF fold change of lived SH-SY5Y GFP cells was significantly observed“ – the change in the number of live target cells was significant

Line 318: “(p < 0.05). While the signal of EthD-1“ – please make one sentence

Line 331: human anti-mouse antibody (HAMA) response

Line 339:” built by joining a single polypeptide chain by flexible linker” – built by joining 2 single-chain antibody fragment sequences

Line 341: “bispecific bivalent scFv platform” – the fragment described is monovalent for each of its specificities, and as it is only one, it is not a platform.

Line 344:” which employe the production” – employed for production

Lines 351-352:” appropriate folding, human-like glycosylation, and post-translation modifications“ – post-translational modifications, and these data would be very enriching for the manuscript. Is the glycosylation or post-translational modifications relevant – as the sequences are not given in the manuscript, it is difficult to judge on this point.

Lines 358-359: “BiTEs exhibited only recognition ability to their targets without any effect on T cell activation and cytotoxicity against neuroblastoma cell line (data not shown)“ – indeed, these data should be shown because here novel reagents are described. As mentioned above, T-cell activation without a target, as well as the cytotoxic effect without the relevant target cells, must be investigated.

Line 359- 360 “…Although the cytotoxic function of BiTEs by redirecting T cells via CD3 is attractive. There are“ – please make one sentence out of these two

Line 361: effect on the immunoregulatory and...

Line 368: “administration of natural and bispecific antibodies” – administration of T-cell specific natural and …

Lines 370-371: “without activation of effector cells“ - without unspecific activation of effector cells

Lines 382-383: word order: “combined CD3/GD2 BiTEs with activated Vγ9Vδ2 T cells“ - CD3/GD2 BiTEs combined with activated Vγ9Vδ2 T cells

Line 386: nM (nanomolar)

Line 391: “to non-treated CD3/GD2 BiTEs” – to tumor cells incubated with effector cells without the presence of CD3/GD2 BiTEs

Line 399: which results in different phenotypes and effector cell characteristics

Line 409: in a format of a CD3/GD2 BiTEs

Line 411: retained the binding reactivity

Line 415: proposes an alternative strategy – it is not provided quite yet

Figure 2. Missing the titration of tumor and effector target-negative cells.

Figure 3. Can you explain why higher concentrations of BiTE were used along with higher numbers of effector cells?

Reviewer #2: The authors present a potentially promising approach to the treatment of recurrent or relapsed neuroblastoma. As others have shown CD3+ T cells, including gamma delta (gd) T cells can be directed to kill tumors cells through engagement of bispecific T cell engagers. The engager used in the present study targets CD3 on T cells and GD2 on neuroblastoma cells. GD2 is the target of a commercial monoclonal antibody, dinutuximab. Overall, the materials used and studies presented are well designed, characterized and analyzed. However, several concerns remain as stated below:

1) It would benefit the study to demonstrate the effectiveness of the approach against multiple neuroblastoma cell lines.

2) Demonstration of in vivo activity in a neuroblastoma tumor model would further benefit the study.

3) The data should be presented as mean +/- standard deviation or better yet, show all of the data points.

4) The gdT cell expansion data should be included. For example, starting cell numbers, final cell numbers, fold-expansion, purity, immunophenotyping.

5) It should be clarified whether or not only one donor was used for gdT expansion. Since donor variability is a major factor in gdT expansion and cytotoxicity, the study would benefit from including gdT cell products from multiple donors.

6) There are a few grammatical error throughout the manuscript that need to be corrected.

Reviewer #3: Immunotherapy with BiTEs and γδT cells are both emerging approaches to cancer therapy, and have shown surprising results when used separately. In this paper, for neuroblastoma (NB), the researchers explored an innovative cancer immunotherapy approach, a bispecific T cell engager (BiTE) against CD3 and GD2, called CD3/GD2 BiTEs, which showed promising results in vitro trials. The results showed that CD3/GD2 BiTEs could effectively bind the target antigen on Jurkat and SH-SY5Y cells in 2D and 3D cell cultures and significantly enhance the cytotoxicity of Vγ9Vδ2 T cells in vitro. This combination therapy could serve as a potential alternative strategy for neuroblastoma treatment, providing new therapeutic ideas for patients with other types of solid tumors.

My additional comments are as follows:

1. No in vivo trials were conducted for CD3/GD2 BiTEs, and the lack of vivo data is a key limitation of this study, which hinders the understanding of the therapeutic potential and safety of CD3/GD2 BiTEs in living organisms. Success in vitro does not always translate to effectiveness in vivo, and due to the lack of animal testing, it is impossible to predict potential toxicity or adverse reactions in complex biological systems.

2. Neuroblastoma has significant genetic and phenotypic heterogeneity, but only SH-SY5Y cells were studied in vitro. Therefore, the efficacy of CD3/GD2 BiTEs on different neuroblastoma subtypes or heterogeneous tumor microenvironments is unclear. This limits the prevalence and effectiveness of CD3/GD2 BiTEs in patients with neuroblastoma.

3. The question of why Vγ9Vδ2 T cells were selected and their advantages over other T cell subpopulations remains to be further discussed. The ability of activated γδT cells to kill a variety of solid tumors and lymphoma cells, including neuroblastoma, is mentioned in the discussion, but specific treatment cases or data should be provided to support this, especially for NB.

4. Although this study shows that CD3/GD2 BiTEs can enhance the cytotoxicity of Vγ9Vδ2 T cells, the mechanism by which CD3/GD2 BiTEs regulates the activity of Vγ9Vδ2 T cells is unclear and can be discussed appropriately.

5. CD3/GD2 BiTEs still retains mouse scFv-GD2. Although its immunogenicity and related reactions triggering HAMA have been discussed, there is still a lack of specific in vivo experimental data verification, which cannot be regarded as a conclusion.

6. Abstract content is not clear and concise, so the purpose, methods, results and conclusions of the abstract can be more clearly divided to improve the readability of the abstract.

7. The richness and aesthetics of the picture content need to be improved, such as adding the action mechanism diagram of CD3/GD2 BiTEs.

8. The citation format of the references is not consistent, some references do not have DOI numbers, etc. Please ensure that the format of all references conforms to the journal guidelines, and some references may need to be adjusted in terms of punctuation, italics or capitalization.

9. There are errors and omissions in some content formats, such as line 210:

%% Dead tumor cells=["dead tatumor cells" (CSFE+ 7-AAD+)/"total tumor cells" (CFE+ 7AAD-) ]×100

6. PLOS authors have the option to publish the peer review history of their article (what does this mean? ). If published, this will include your full peer review and any attached files.

**Do you want your identity to be public for this peer review?** For information about this choice, including consent withdrawal, please see our Privacy Policy .

Reviewer #1: No

Reviewer #2: No

Reviewer #3: No

---

## [Author Response · Author response to Decision Letter 1]

3 Feb 2025

Dear reviewers,

We would like to thank you for sending comments following the peer review of our manuscript.

We are very appreciating the interest that reviewers have taken in our manuscript. We would like to gratefully thank the reviewers for providing valuable comments which helped to improve our manuscript markedly. In the revised version, we have addressed concern of the reviewers. Changes to the text are highlighted in yellow.

We do hope that the revised version of our study is now suitable to publish in PLOS ONE.

Thank you for your consideration of our revised manuscript.

Yours sincerely,

Suradej Hongeng, MD

---

## [Decision Letter · Decision Letter 1]

9 Mar 2025

PONE-D-24-27594R1Combining CD3/GD2 bispecific T cell engager with human Vγ9Vδ2 T cells facilitates neuroblastoma cell targeting and killing in vitroPLOS ONE

Dear Dr. hongeng,

Thank you for submitting your manuscript to PLOS ONE. After careful consideration, we feel that it has merit but does not fully meet PLOS ONE’s publication criteria as it currently stands. Therefore, we invite you to submit a revised version of the manuscript that addresses the points raised during the review process. The reviewers have provided additional suggestions to improve the manuscript, which can be found at the bottom of this email. Please address the raised questions one by one, and incorporate the suggestions clearly.

We look forward to receiving your revised manuscript.

Kind regards,

Dr Mahmood S Choudhery, PhD

Academic Editor

PLOS ONE

Reviewers' comments:

Reviewer's Responses to Questions

**Comments to the Author**

1. If the authors have adequately addressed your comments raised in a previous round of review and you feel that this manuscript is now acceptable for publication, you may indicate that here to bypass the “Comments to the Author” section, enter your conflict of interest statement in the “Confidential to Editor” section, and submit your "Accept" recommendation.

Reviewer #4: (No Response)

Reviewer #5: All comments have been addressed

2. Is the manuscript technically sound, and do the data support the conclusions?

Reviewer #4: Partly

Reviewer #5: Partly

3. Has the statistical analysis been performed appropriately and rigorously? 

Reviewer #4: Yes

Reviewer #5: Yes

4. Have the authors made all data underlying the findings in their manuscript fully available?

Reviewer #4: Yes

Reviewer #5: Yes

5. Is the manuscript presented in an intelligible fashion and written in standard English?

Reviewer #4: Yes

Reviewer #5: Yes

6. Review Comments to the Author

Reviewer #4: Below are the main questions and suggestions.

1. The main flaw of the article is the lack of necessary in vivo experiments to verify the anti-tumor efficacy of CD3/GD2 bispecific T cell engager because in vitro experiments often fail to achieve consistent results in animal models due to overly simple systems.

2. As the author mentioned, this bispecific antibody was composed of human scFv CD3 and mouse scFv GD2. Where does the sequence of scFv GD2 come from? If affinity screening and humanization modification can be made, it will be of great significance for future clinical applications.

3. In Fig 3, the author verified the specific binding between CD3/GD2 BiTE and CD3 positive Jurkat cells. How about primary expanded Vγ9Vδ2 T cells in PBMC? Is it only a specific binding between CD3/GD2 BiTE and Vγ9Vδ2 T cells, or will it transmit activation signals, leading to activation and proliferation of Vγ9Vδ2 T cells? Besides activation and proliferation, activated Vγ9Vδ2 T cells also secrete effector cytokines such as IFN-γ. All of these are very important to prove the anti-tumor functions of CD3/GD2 BiTE.

4. Functional verification is the most important part of the article. But the entire article only used one GD2 positive cell line SH-SY5Y. Other more target cells with GD2 expression should be used to repeatedly validate cytotoxic activity results. Gran B and perforin that are key molecules in cytotoxicity must be analyzed.

5. We all know that CD3 is not unique to Vγ9Vδ2 T cells, so can CD3/GD2 BiTE also promote the anti-tumor function of αβT cells in vitro? Suggest the author to conduct comparative experiments to illustrate the problem.

6. All the figures are not well qualified and need to be improved.

Reviewer #5: Ref: PONE-D-24-27594R1

Title: Combining CD3/GD2 bispecific T cell engager with human Vγ9Vδ2 T cells facilitates

neuroblastoma cell targeting and killing in vitro

Reviewers' comments:

The authors have made detailed modifications and corrections in response all comments, addressing most issues adequately. However, there remain some areas of concern:

1. The most important deficiency of the study is lacking verifications in vivo.

2. No evaluation on the effectiveness across a broader range of neuroblastoma subtypes.

3. Further discussion on the rationale for choosing Vγ9Vδ2 T cells is needed.

7. PLOS authors have the option to publish the peer review history of their article (what does this mean? ). If published, this will include your full peer review and any attached files.

**Do you want your identity to be public for this peer review?** For information about this choice, including consent withdrawal, please see our Privacy Policy .

Reviewer #4: No

Reviewer #5: **Yes: ** Zhe Cai

---

## [Author Response · Author response to Decision Letter 2]

25 Mar 2025

Dear reviewers,

We would like to send a point-by-point response to the reviewer’s comments.

In the revised manuscript, we have addressed the concern of the reviewers. Changes to the text are highlighted in yellow.

We do hope that the second revised version of our study is now understandable and suitable to publish in PLOS ONE.

Thank you for your consideration of our revised manuscript.

Yours sincerely,

Suradej Hongeng, MD

---

## [Decision Letter · Decision Letter 2]

8 Apr 2025

PONE-D-24-27594R2Combining CD3/GD2 bispecific T cell engager with human Vγ9Vδ2 T cells facilitates neuroblastoma cell targeting and killing in vitroPLOS ONE

Dear Dr. hongeng, Thank you for submitting your manuscript to PLOS ONE. After careful consideration, we feel that it has merit but does not fully meet PLOS ONE’s publication criteria as it currently stands. Therefore, we invite you to submit a revised version of the manuscript that addresses the points raised during the review process. The reviewer requires clarification and the incorporation of necessary changes in the manuscript to improve its overall quality.  Please submit your revised manuscript by May 23 2025 11:59PM. If you will need more time than this to complete your revisions, please reply to this message or contact the journal office at plosone@plos.org . Please include the following items when submitting your revised manuscript:

We look forward to receiving your revised manuscript.

Kind regards,

Dr Mahmood S Choudhery, PhD

Academic Editor

PLOS ONE

Journal Requirements:

Reviewers' comments:

Reviewer's Responses to Questions

**Comments to the Author**

1. If the authors have adequately addressed your comments raised in a previous round of review and you feel that this manuscript is now acceptable for publication, you may indicate that here to bypass the “Comments to the Author” section, enter your conflict of interest statement in the “Confidential to Editor” section, and submit your "Accept" recommendation.

Reviewer #4: (No Response)

Reviewer #5: All comments have been addressed

2. Is the manuscript technically sound, and do the data support the conclusions?

Reviewer #4: Yes

Reviewer #5: Yes

3. Has the statistical analysis been performed appropriately and rigorously? 

Reviewer #4: Yes

Reviewer #5: Yes

4. Have the authors made all data underlying the findings in their manuscript fully available?

Reviewer #4: Yes

Reviewer #5: Yes

5. Is the manuscript presented in an intelligible fashion and written in standard English?

Reviewer #4: Yes

Reviewer #5: Yes

6. Review Comments to the Author

Reviewer #4: The author addressed that CD3/GD2 BiTE only recognized the CD3 molecule but did not activate T cells. I still insist that an effective anti-tumor BiTE should be able to activate anti-tumor effector cells in vivo, not just as a target. If an anti-CD3 scFv sequence that can both target and activate CD3+ T cells could be selected during screening, it will have a more promising application prospect.

In addition, the author conducted experiments using PBMCs and did not observe an increase in anti-tumor cytotoxicity. So what about activated T cells using CD3/CD28 beads? If the author emphasizes that their BiTE can only bind to activated V γ 9V δ 2 T cells and enhance the targeting to GD2+ tumor cells, a scFv targeting anti-V γ 9V δ 2 TCR should be screened instead of targeting CD3 molecules.

Since the anti-tumor efficacy in vivo cannot be achieved, it is necessary to verify using several different target cells and effector cells in vitro. However, I did not see the authors make every effort to improve it in the revised manuscript.

Reviewer #5: The authors addressed all of my concerns. Their answer is reasonable. And there is no more comments can be listed now.

7. PLOS authors have the option to publish the peer review history of their article (what does this mean? ). If published, this will include your full peer review and any attached files.

**Do you want your identity to be public for this peer review?** For information about this choice, including consent withdrawal, please see our Privacy Policy .

Reviewer #4: No

Reviewer #5: **Yes: ** Zhe Cai

---

## [Author Response · Author response to Decision Letter 3]

2 May 2025

Dear reviewer,

We would like to thank you for your valuable comments and suggestions. We have addressed your concerns and here are the responses regarding your comments.

We do hope that this revised version of our study is suitable to publish in PLOS ONE.

Thank you for your consideration of our revised manuscript.

Yours sincerely,

Suradej Hongeng, MD

---

## [Editor Report · Decision Letter 3]

13 May 2025

Combining CD3/GD2 bispecific T cell engager with human Vγ9Vδ2 T cells facilitates neuroblastoma cell targeting and killing in vitro

PONE-D-24-27594R3

Dear Dr. Hongeng,

We’re pleased to inform you that your manuscript has been judged scientifically suitable for publication and will be formally accepted for publication once it meets all outstanding technical requirements.

Kind regards,

Dr Mahmood S Choudhery, PhD

Academic Editor

PLOS ONE
---

## [Editor Report · Acceptance letter]

PONE-D-24-27594R3

PLOS ONE

Dear Dr. Hongeng,

I'm pleased to inform you that your manuscript has been deemed suitable for publication in PLOS ONE. Congratulations! Your manuscript is now being handed over to our production team.

Kind regards,

on behalf of

Dr. Mahmood S Choudhery

Academic Editor

PLOS ONE